**Subject Category:**
Biology (whole organism)

ecology/health and disease and epidemiology

species richness – disease relationship,
generalized additive model, the Third Pandemic

**Authors for correspondence:**
Jianguo Xu
e-mail: xujianguo@icdc.cn
Nils Chr. Stenseth
e-mail: n.c.stenseth@ibv.uio.no
Bing Xu
e-mail: bingxu@tsinghua.edu.cn

†Z.S. and L.X. contributed equally to this work.

# Human plague system associated with rodent diversity and other environmental factors

Zhe Sun[1,2,3,†], Lei Xu[1,2,3,4,†], Boris V. Schmid[2], Katharine R. Dean[2], Zhibin Zhang[5], Yan Xie[5], Xiye Fang[4], Shuchun Wang[4], Qiyong Liu[4], Baolei Lyu[1], Xinru Wan[5], Jianguo Xu[4], Nils Chr. Stenseth[1,2,3] and Bing Xu[1,3]

[1]Ministry of Education Key Laboratory for Earth System Modeling, Department of Earth System Science, Tsinghua University, Beijing 100084, People's Republic of China
[2]Centre for Ecological and Evolutionary Synthesis (CEES), Department of Biosciences, University of Oslo, N-0316 Oslo, Norway
[3]Joint Center for Global Change Studies, Beijing 100875, People's Republic of China
[4]State Key Laboratory of Infectious Disease Prevention and Control, National Institute for Communicable Disease Control and Prevention, Chinese Center for Disease Control and Prevention, Changping, Beijing 102206, People's Republic of China
[5]State Key Laboratory of Integrated Management on Pest Insects and Rodents, Institute of Zoology, Chinese Academy of Sciences, Beijing 100101, People's Republic of China

 ZS, 0000-0003-1200-8141; BVS, 0000-0003-0452-623X;
ZZ, 0000-0003-2090-7999; NCS, 0000-0002-1591-5399

Plague remains a threat to public health and is considered as a re-emerging infectious disease today. Rodents play an important role as major hosts in plague persistence and driving plague outbreaks in natural foci; however, few studies have tested the association between host diversity in ecosystems and human plague risk. Here we use zero-inflated generalized additive models to examine the association of species richness with human plague presence (where plague outbreaks could occur) and intensity (the average number of annual human cases when they occurred) in China during the Third Pandemic. We also account for transportation network density, annual precipitation levels and human population size. We found rodent species richness, particularly of rodent plague hosts, is positively associated with the presence of human plague. Further investigation shows that species richness of both wild and commensal rodent plague hosts are positively correlated with the presence, but only the latter correlated with the intensity. Our results indicated a positive relationship between rodent diversity and human plague, which may provide suggestions for the plague surveillance system.

# 1. Introduction

Plague is a vector-borne disease caused by *Yersinia pestis*. The natural cycle of plague involves rodents as hosts and associated fleas as vectors [1]. Plague is primarily a zoonosis, and many species of rodents are susceptible to the infection and can transmit the disease [2]. Other terrestrial vertebrates, including mammals and birds, can also be infected with plague [1]. Humans can acquire plague both from plague hosts and other humans, indirectly through fleabites or directly by contact with infectious droplets and tissues [3,4].

There have been three major plague pandemics in human history. The First Pandemic, beginning with the Justinianic Plague, and the Second Pandemic, beginning with the Black Death, killed a significant proportion of the population in North Africa and Europe [5,6]. The Third Pandemic originated in the Yunnan province in the southwest part of China in 1772 and then spread along the southeast coast of China towards Hong Kong [7]. Plague outbreaks also appeared in northern China in 1854, but it is unclear if these derived from the epidemics in the south [8]. When the Third Pandemic reached Hong Kong in 1894, it spread globally via maritime transportation [9]. Today, plague remains a threat to public health and is considered a re-emerging disease due to sporadic outbreaks in Africa, Asia and the Americas [10,11].

Abiotic and anthropogenic factors, such as annual precipitation levels, local human population size and transportation connectivity (main transportation network density) are known to affect through various mechanisms where local plague outbreaks occur, the intensity of those outbreaks and further dissemination of the disease [8,9,12]. Precipitation levels affect plague transmission by influencing the rodent and flea population densities [12–14], which in turn affect the spillover chance of plague into human populations [12,15]. Local human population size has a nonlinear relationship to the degree of interaction that the population will have with wildlife, and thus with the risk of contracting plague from a wildlife reservoir; but when plague outbreaks occur in larger human populations, they, in general, result in larger outbreaks of the disease, either directly (in the case of human-to-human transmission) or indirectly (in the case of rodent-borne plague outbreaks, as the number of commensal rodents is expected to be larger in locales with larger human populations) [6,9,10]. Transportation routes, a factor partially correlated to human population density, can reshape the distribution of plague by accelerating the spread of infected animals and humans to a larger area, thus affecting both the presence and the intensity of outbreaks [8,16]. Previous studies have demonstrated that transportation accounted for the introduction and subsequent spread of plague in Europe [17,18], as well as the dispersion of plague in China [8].

Variations in rodent and flea population densities within wildlife plague systems have been intensively studied, as they are known to affect the prevalence of plague in wildlife populations and the spillover risk to humans [15,19,20]. Less studied, however, is the general impact of host diversity in ecosystems on human plague risk [21,22]. Host diversity can be measured by species richness (the total number of unique species present), or more informative, species evenness (a combination of species richness and their relative abundance) [21]. Often the only approximate measure available over large geographical regions is species richness. The human infectious diseases are affected by both the local innate species richness and their changes [23–27]. Regions with high species richness are likely to be source pools for new pathogens [25]. Species richness has been shown to have a positive association with the emergence of zoonotic diseases [24,27] and the species richness of human pathogens [26]. High innate species richness may provide a stable system for pathogen maintenance and transmission and also prevent local extinction of pathogens, which in turn affects human health [26]. However, the loss of biodiversity frequently leads to increases in infectious diseases, known as the dilution effect [23]. Dilution effect often occurs in disease systems in which hosts differ in competence for transmission of pathogens and the competent host populations increase as the biodiversity declines [28]. Instead, high diversity with non-competent hosts tends to lower the abundance of competent hosts and reduces the disease risk [23,28].

A large part of China's geographical territory contains plague foci [29], where plague is still endemic today among the rodent populations [14]. High spatial heterogeneity of biotic and other environmental factors has led to variability in plague risk (figure 1*a*) in China. Considering the unknown effect of innate host species richness (figure 2) on plague, our study focused on this biotic factor, together with average precipitation levels, human population size and local transportation connectivity (figure 1*b*), three other factors known to be of importance in regulating plague risk [8,9]. Based on a dataset of high-resolution village-level human plague data during the Third Pandemic (figure 1; electronic supplementary material, figure S1), we analysed the association of these four factors with where human plague outbreaks could occur (presence) and the average

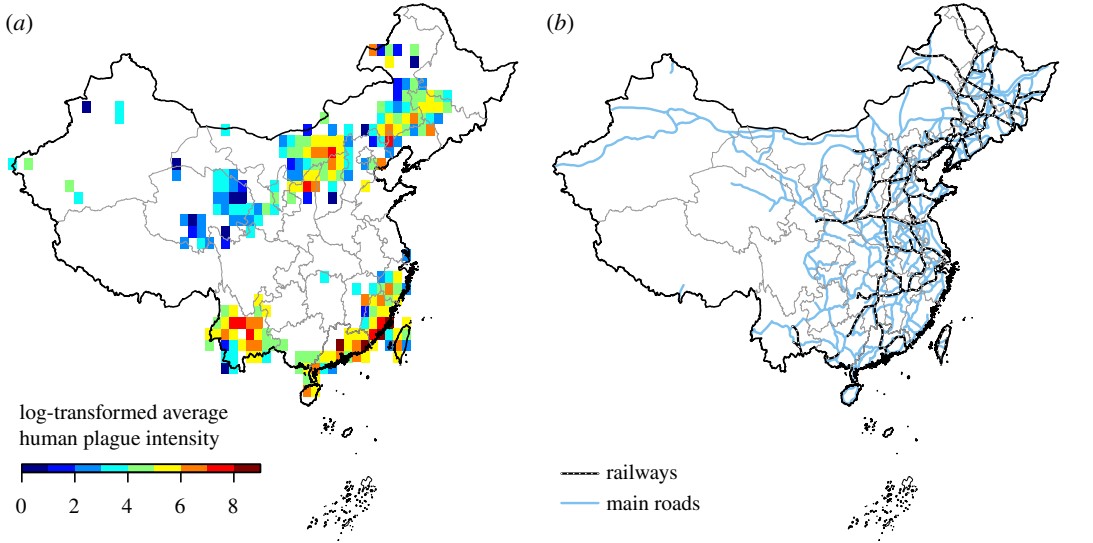

**Figure 1.** Human plague intensity in mainland China from 1772 to 1964 and transportation routes. (*a*) Average human plague intensity (average number of annual human cases when plague occurred) after logarithmic transformation in 1° grids in mainland China. Colours show grids with presence of plague. (*b*) Historical railways and main roads in mainland China.

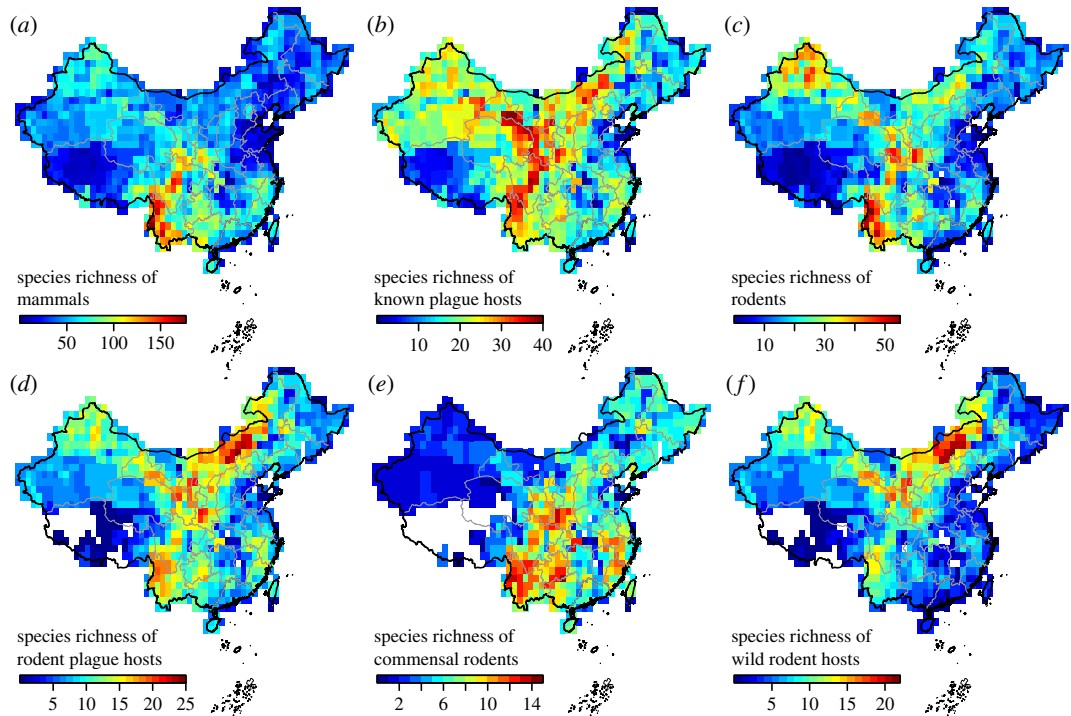

**Figure 2.** Spatial distribution of host diversity in mainland China. Species richness of (*a*) all mammals, (*b*) known plague reservoir hosts, (*c*) all rodents, (*d*) rodent plague hosts, (*e*) commensal rodent plague hosts, and (*f*) wild rodent plague hosts.

number of annual human cases when they occurred (intensity) by using zero-inflated generalized additive models (ZIGAMs) [30].

# 2. Material and methods

## 2.1. Historical plague and human population data

In 1963, the Health and Epidemic Prevention Bureau, Ministry of Health started organizing the Institute of Epidemiology and Microbiology, Chinese Academy of Medical Sciences and local plague control units

to collect and compile records of historical human plague in China. Detailed spatio-temporal information on outbreaks (including years and locations) and intensity (number of cases) of plague from 1772 to 1964 on a village-level basis were documented in the dataset [31]. We subsequently aggregated the number of cases on a 1° basis. We further defined the plague presence/absence in grids as a binary variable, where 1 represents plague could occur and 0 represents plague never occurs. The first appearance of plague differed among regions [8], and plague outbreaks were often discontinuous, with years of stasis in recorded cases [31], which does not allow for proper incidence measurement (average number of cases per year across the study period). Thus, we calculated the average plague intensity ($P_i$) in grid cell $i$ based on the total number of historical plague cases ($C_i$) and the number of years with plague in grid cell $i$ ($N_i$), according to the following formula,

$$P_i = \begin{cases} 0, & C_i = 0 \\ \dfrac{C_i}{N_i}, & C_i > 0 \end{cases}.$$

(2.1)

We obtained 5 min gridded population data with a time interval of 10 years from Goldewijk *et al.* [32]. Assuming that the demographic change was roughly linear during each 10 year time interval, we calculated the population in each year and aggregated it on a 1° basis. Long-term population size in grids was the average population during years with plague or average population from 1772 to 1964 for grids without plague cases (electronic supplementary material, text).

## 2.2. Species richness data

County-level species distributions of terrestrial vertebrates (mammals and birds) were extracted from a database compiled by Xie *et al.* [33]. By integrating relevant information in the literature, this database is an aggregation of species occurrence from different regions and historical periods (beginning in 1758, see electronic supplementary material, figure S2). The timespan of the dataset approximately covers the Third Pandemic in China. We used the whole dataset of species distributions to calculate species richness, to ensure the spatial integrity. The correlation test between vertebrate species richness of the whole dataset and subsets from different periods, as well as a national animal distribution survey [34], showed that the species distribution pattern from different periods was relatively stable (electronic supplementary material, table S1).

There are 1797 species of terrestrial vertebrates recorded in the database, which potentially serve as plague hosts or play an indirect role in plague through predation or competition relations with plague hosts [1,35]. We started from this assemblage, consistent with previous studies of diversity–disease relationship [26,36]. Among these 1797 species, there are 563 species of mammals. As plague is primarily a disease of rodents [2], we also filtered 191 rodent species from the database. Additionally, there are 88 species of known plague hosts in China which are involved in plague ecosystems and transmission ('plague hosts' for short) based on field surveys of naturally infected animals and aetiological studies (electronic supplementary material, text), with 85 species in Mammalia and 3 species in Aves [35]. Of these 88 species, 54 species are rodent plague hosts. The rodent plague hosts were further divided into two subsets, commensal (living in house or farmland) and wild rodents (living in prairie, desert and forest) according to their habitat (electronic supplementary material, table S2). We overlaid the county-level spatial distribution of each species (polygons) and the 1° grids covering China and determined the presence or absence of a specific species in a grid by 1 or 0. Finally, we calculated separately the gridded species richness of terrestrial vertebrates, mammals, rodents, known plague hosts, rodent plague hosts, commensal rodent plague hosts and wild rodent plague hosts based on the sum of species' presence/absence in grids in each category. We also calculated the above-mentioned species richness based on the species distribution records before 1964 (electronic supplementary material, figure S3), for testing the robustness of our results.

## 2.3. Transportation and precipitation

Railways and roads constituted primary overland transportation modes for the study period during the Third Pandemic in China. We digitized a historical map from the first half of the twentieth century to obtain railways and main roads (figure 1b) [37]. Traffic flow is in general considered crucial for shaping the distribution and prevalence of infectious diseases [38]. However, historical information of traffic flow was not available. Therefore, we used the total length of railways and roads in 1° grids as a measurement of transportation connectivity in this study. We also tested the robustness of the

transportation effect by using transportation routes of the nineteenth century (electronic supplementary material, text).

Long-term monthly precipitation records from 1960 to 1990 were acquired from WorldClim (http://www.worldclim.org/) and converted to annual precipitation. The precipitation was standardized around a mean 0 with a standard deviation of 1.

## 2.4. Zero-inflated generalized additive model

As time-series data of species richness and transportation connectivity were not available, we used a spatial generalized additive model to analyse the association between human plague and environmental factors, in order to avoid the pseudo-replication problem [39] caused by including every grid cell in every year from 1772 to 1964. The average plague intensity in China is zero-inflated (electronic supplementary material, figure S1). Thus, we used ZIGAMs [30] to analyse the association of host species richness, precipitation, human population and transportation connectivity with both human plague presence and average intensity in China. We used the ZIGAM functions developed by Liu & Chan [30] in R language. Our ZIGAM model is composed of two parts: a binomial part depicting the presence (1) or absence (0) of plague in grids, and a lognormal part analysing the positive plague intensity. The initial model is as follows,

$$P_i = B\big[\alpha + f_1(Ver_i) + f_2(Prec_i) + f_3(Popu_i) + f_4(Trans_i)\big] \times \exp\big[\beta + f_5(Ver_i) + f_6(Prec_i) \\ + f_7(Popu_i) + f_8(Trans_i) + \varepsilon_i\big].$$

(2.2)

Here, $P_i$ is the average plague intensity in grid cell $i$. $B(\cdot)$ is a logit link function depicting the presence/absence of plague. The parameters $\alpha$ and $\beta$ are the intercepts. The functions $f_1, f_2, f_3, \ldots, f_8$ are smooth functions (cubic regression spline lines), all with maximally 2 degrees of freedom. $Ver_i$ is the species richness of terrestrial vertebrates in grid $i$. $Prec_i$ is the annual precipitation after standardization. $Popu_i$ and $Trans_i$ are log-transformed human population size and log-transformed transportation connectivity (plus 1 to avoid zero logarithm) in grid $i$, respectively. $\varepsilon_i$ is the random error.

We used a backward model selection procedure to elect explanatory variables [40]. The logarithm of marginal likelihood (logE) was used for the model selection and a model with larger logE indicated a better model with maximum posterior model probability [30]. We compared models with different measures of species richness by replacing the species richness of terrestrial vertebrates with species richness of mammals, rodents, known plague hosts, rodent plague hosts, commensal rodent plague hosts and wild rodent plague hosts, using logE, and listed the amount of variance explained for both the presence and the intensity of plague outbreaks. We tested the spatial correlations of residuals in the lognormal part in the models by using semi-variogram [8]. Interaction of species richness and transportation was also tested to check if there were any interaction effects [41]. For the subset of vertebrate species richness with the largest logE, we classified the grid cells into two categories, high-diversity and low-diversity groups, by using the average value of species richness in China as a threshold. We then tested the association strength and significance between species richness and human plague (electronic supplementary material, text) in these two groups using risk ratio (RR) [42] and *chi-square* test [43].

## 3. Results

We explored different subsets of vertebrate species richness (figure 2; electronic supplementary material, figure S3); in addition to considering the local species richness of all 1797 terrestrial vertebrate species in the biodiversity dataset in model (*i*), we considered models that only looked at the subset of 563 mammal species (*ii*), 191 rodent species (*iii*), 88 species known to be involved in plague ecosystems and transmission in China (*iv*), 54 species of rodent plague hosts (*v*), and of those in only the subset that are wild rodent plague hosts (*vi*), and only the subset that are commensal rodent plague hosts (*vii*). Finally, we also considered a baseline model (*viii*) that did not include diversity at all (table 1).

Using backward model selection, we did not find significant associations between the species richness of terrestrial vertebrates and plague presence or average intensity (model (*i*)). The species richness of mammals and known plague hosts were all in a positive relationship with the plague presence, but not significant with plague intensity (electronic supplementary material, text).

**Table 1.** ZIGAM results of models with and without host diversity.

| rank | model type | model formula | logE | variance explained (presence) | variance explained (intensity) |
|---|---|---|---|---|---|
| 1 | (v) with species richness of known rodent plague hosts | $P_i = B[\alpha + f_{15}(Rh_i) + f_2(Prec_i) + f_3(Popu_i) + f_4(Trans_i)]$ $\times \exp[\beta + f_{16}(Rh_i) + f_6(Prec_i) + f_7(Popu_i) + f_8(Trans_i) + \varepsilon_i]$ | −869.33[a] | 19.2% | 26.3% |
| 2 | (vi) with species richness of wild rodent plague hosts | $P_i = B[\alpha + f_{17}(Wild_i) + f_2(Prec_i) + f_3(Popu_i) + f_4(Trans_i)]$ $\times \exp[\beta + f_{18}(Wild_i) + f_6(Prec_i) + f_7(Popu_i) + f_8(Trans_i) + \varepsilon_i]$ | −871.53 | 19.0% | 24.9% |
| 3 | (vii) with species richness of commensal rodent plague hosts | $P_i = B[\alpha + f_{19}(Com_i) + f_2(Prec_i) + f_3(Popu_i)]$ $\times \exp[\beta + f_{20}(Com_i) + f_6(Prec_i) + f_7(Popu_i) + \varepsilon_i]$ | −883.13 | 16.4% | 25.3% |
| 4 | (iii) with species richness of rodents | $P_i = B[\alpha + f_{11}(Rod_i) + f_2(Prec_i) + f_3(Popu_i) + f_4(Trans_i)]$ $\times \exp[\beta + f_{12}(Rod_i) + f_6(Prec_i) + f_7(Popu_i) + f_8(Trans_i) + \varepsilon_i]$ | −883.96 | 16.7% | 24.8% |
| 5 | (iv) with species richness of known plague hosts | $P_i = B[\alpha + f_{13}(Host_i) + f_2(Prec_i) + f_3(Popu_i) + f_4(Trans_i)]$ $\times \exp[\beta + f_{14}(Host_i) + f_6(Prec_i) + f_7(Popu_i) + f_8(Trans_i) + \varepsilon_i]$ | −884.89 | 17.8% | 25.0% |
| 6 | (ii) with species richness of mammals | $P_i = B[\alpha + f_9(Mam_i) + f_2(Prec_i) + f_3(Popu_i) + f_4(Trans_i)]$ $\times \exp[\beta + f_{10}(Mam_i) + f_6(Prec_i) + f_7(Popu_i) + f_8(Trans_i) + \varepsilon_i]$ | −890.46 | 15.5% | 24.3% |
| 7 | (i) with species richness of terrestrial vertebrates | $P_i = B[\alpha + f_1(Ver_i) + f_3(Popu_i) + f_4(Trans_i)]$ $\times \exp[\beta + f_5(Ver_i) + f_7(Popu_i) + f_8(Trans_i) + \varepsilon_i]$ | −894.88 | 14.4% | 23.0% |
| 8 | (viii) no species richness | $P_i = B[\alpha + f_2(Prec_i) + f_3(Popu_i) + f_4(Trans_i)]$ $\times \exp[\beta + f_6(Prec_i) + f_7(Popu_i) + f_8(Trans_i) + \varepsilon_i]$ | −898.48 | 14.9% | 24.1% |

[a]Final model with the largest logE.

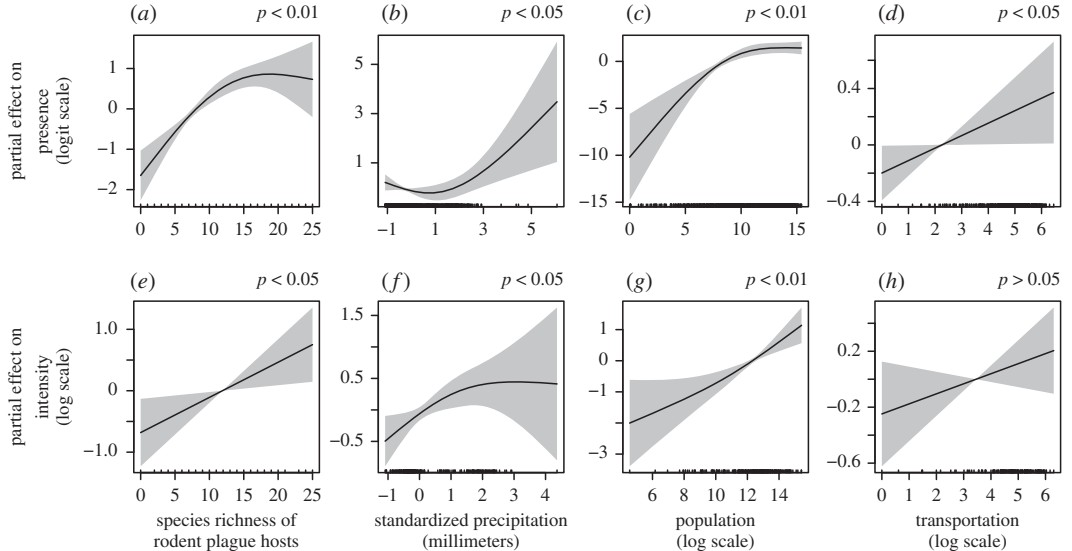

**Figure 3.** Partial effects of smooth functions on human plague presence and intensity of the model with species richness of rodent plague hosts. (*a–d*) The binomial part quantifies the presence of plague. (*e–h*) The lognormal part evaluates the positive plague intensity.

Delving more in-depth into the models with diversity related to the rodents, we found the best-fitting model variant for predicting where plague outbreaks could occur, and their intensity was model (*v*), which considered the local species richness of rodent plague hosts. Model (*v*) gave a marginal improvement of the model performance, with a 1.5–4.3% point increase in the variance explained of the presence of plague, to a maximum of 19.2%, and a 0.7–2.2% point increase to a maximum of 26.3% of the variance explained of the intensity of plague outbreaks (table 1). Using a 1° spatial grid covering China, we found that the local species richness of rodent plague hosts was positively associated with both plague presence ($\chi^2_{1.88,7.69} = 39.28$; $p < 0.01$; figure 3*a*) and the intensity in humans ($F_{1.00,5.84} = 6.16$; $p < 0.05$; figure 3*e*). In this model, the annual precipitation levels had a positive association with both plague presence ($\chi^2_{1.88,7.69} = 8.33$; $p < 0.05$; figure 3*b*) and intensity ($F_{1.46,5.84} = 4.35$; $p < 0.05$; figure 3*f*), as did local human population size (for presence: $\chi^2_{1.93,7.69} = 28.36$; $p < 0.01$; figure 3*c*; for intensity: $F_{1.38,5.84} = 11.42$; $p < 0.01$; figure 3*g*). The local transportation connectivity showed a positive relationship for plague presence ($\chi^2_{1.00,7.69} = 4.24$; $p < 0.05$; figure 3*d*) and a non-significant positive association with plague intensity ($p > 0.05$; figure 3*h*). After classifying the grid cells into high-diversity and low-diversity groups, we conducted a *chi-square* test and calculated *RR* (electronic supplementary material, text; table S3 and table S4). A significant association was found between high diversity and human plague presence ($\chi^2 = 60.88$; d.f. = 1; $p < 0.01$). Besides, *RR* was 2.51 (95% CI: 1.98–3.20; $p < 0.01$), indicating that grids with a high diversity of rodent plague hosts had 2.51 times the risk of plague presence compared to grids with low diversity.

The residuals in the model showed some marginal spatial correlation between directly adjacent grid cells (electronic supplementary material, figure S4). In a sensitivity analysis where we merged our grid cells to 2° × 2°, we found the species richness of rodent hosts was still the best predictor of human plague risk (in both presence and intensity). In this coarser-scale model, the species richness of rodent plague hosts was still positively associated with both plague presence ($\chi^2_{1.00,5.17} = 13.88$; $p < 0.01$) and intensity ($F_{1.00,5.00} = 5.62$; $p < 0.05$). At a 2° × 2° resolution, annual precipitation, local human population size and transportation connectivity showed instability in their significant relationships with plague presence and intensity but largely kept the same relationships as represented in figure 3 (electronic supplementary material, figure S5).

In models with different subsets of rodent plague hosts, (*vi*) (wild rodent plague hosts) and (*vii*) (commensal rodent plague hosts), we see that the species richness of wild rodent hosts showed a significant positive association with plague presence, but not with human plague intensity (electronic supplementary material, figure S6). The species richness of commensal rodent hosts, however, had a positive association with both human plague presence and intensity (electronic supplementary material, figure S6), suggesting that it is the composition of the commensal rodent population that is of particular importance for the human average intensity of plague.

# 4. Discussion

Our results revealed a positive link between the rodent diversity and human plague in China during the Third Pandemic. Our analysis provides evidence that the increased species richness of rodent plague hosts is in a positive association with both human plague presence and the intensity. Further investigation shows that species richness of both wild and commensal rodent plague hosts are positively correlated with the presence, but only the latter correlated with the intensity.

## 4.1. Association with species richness

The presence and intensity of human plague had a significant positive association with the species richness of rodent plague hosts in China. Our result was consistent with previous studies [44,45]. Eisen *et al.* [44] reported that the plague presence in humans increased in the Rocky Mountain/Great Basin regions of the United States and attributed the results to the high diversity of rodents in these areas. Bonvicino *et al.* [45] found that rodent species richness was higher in the natural plague foci in South America. The positive species richness–human plague relationship may result from an association between high species richness of hosts and the maintenance of *Y. pestis* [22]. High species richness of rodent plague hosts increases the odds that there is a resistant species which is able to provide refuge to *Y. pestis* and prevent the pathogen from local extinction [26,46]. High species richness of rodent plague hosts increases the possibility of pathogen maintenance in the natural foci [45], which in turn increases spillover chance of plague into human populations. Additionally, multispecies interaction in species-rich regions may accelerate the evolution of different variants of *Y. pestis*, which contributes to the adaptation of the pathogen to hosts and new niches in regions [47]. The microevolution that accompanies the transmission and expansion of *Y. pestis* may help the pathogen persist in new environments, form plague foci and increase the human plague presence [48].

The positive association between rodent host diversity and human plague may also relate to the species composition of wild and commensal rodents. We found high species richness of wild rodent plague hosts was associated with a high possibility of plague presence, but not significant with plague intensity. This can be explained due to the fact that wild rodents are the natural reservoirs for plague, but typically live in remote areas away from humans [49]. High species richness of wild rodents increases the odds of a persistent plague transmission cycle in the natural environment [45]. For instance, the presence of grasshopper mouse (*Onychomys leucogaster*) increased plague persistence and outbreaks in prairie dog (*Cynomys ludovicianus*) populations, the primary plague host species in the western United States [50]. However, spillover events directly from wild rodents to humans depend on wild rodents entering human settlements or human activities that encroach on wildlife, such as hunting or farming [31,51]. Transmission of plague from wild rodents to human has been shown to involve domestic animals as an intermediary in Yunnan Province in China [52]. We found species richness of commensal rodent plague hosts was associated with both human plague presence and intensity. This is probably because commensal rodents tend to have a high contact rate with both wild rodents and humans, and thus the local presence of one or more commensal rodent plague host species makes it more likely that a bridge between the local wild rodent plague reservoir and human populations can be formed [10]. Once commensal rodents are infected with plague, the odds of human plague presence therefore increases [3,53]. In addition, commensal rodent plague hosts might bring an epizootic into an urban environment and thus create more plague-infected fleas that can infect the domestic animals and humans, increasing the transmission potential and amplifying human plague intensity [15]. One thing worth noting is that our estimates of species richness are the maximum species occurrence during the study period, which are derived from the best available records but not from the field estimates. However, the correlation test among species richness from this dataset, subsets of this dataset and a national animal distribution survey revealed that the species distribution pattern was relatively stable and thus our estimates provided species richness pattern for reference.

## 4.2. Human plague risk near the Heihe-Tengchong Line

The Heihe-Tengchong Line is a demarcation based on the distribution of human population in China [54]. We observed a clear dividing line in the spatial distribution of the ratios of wild/commensal rodent plague hosts to all rodent plague hosts (electronic supplementary material, figure S7) in China

that also separates on the Heihe-Tengchong Line. The west side of the line is sparsely populated by people and has a high ratio of wild rodents. On the contrary, the east side of the line is densely populated and has a large proportion of commensal rodents. Regions near the line consist of farming-pastoral ecozones where a high diversity of wild rodents and commensal rodents blend and interact, and according to our results would therefore be at higher risk of plague. However, the central part of China, which is intersected by the Heihe-Tengchong Line, appears to have been free of plague during the Third Pandemic (electronic supplementary material, figure S7C). However, there are historical records of plague outbreaks in this region, like the thirteenth-century outbreak in Bianjing [55], which are speculative in nature. Other factors that modulated the risk of plague in Central China might therefore play a role that we are not aware of yet.

## 4.3. Association with human population and transportation connectivity

We found a positive association between local human population size and plague, which was consistent with previous studies [9,18]. Population size plays a role in the plague as large local human population size increases the contact rate among humans and human-to-human transmission potential [6,9]. Additionally, regions with a large local human population tend to be more developed in social resources such as transportation [18], which contribute to the diffusion of plague and are discussed in more detail in below.

The positive association between transportation connectivity and plague presence was consistent with previous studies that have shown transportation can accelerate the spread of plague [8,18]. Transportation routes promoted travel of goods and people and related to large-scale human population movements and migration. In return, these overland routes increased the potential of transmitting plague-infected rats, fleas and patients to a larger area. The trade routes from Dali to Kunming were believed to account for the spread of bubonic plague from western to eastern Yunnan at the beginning of the Third Pandemic in China [16]. During the 1910–1911 outbreak of pneumonic plague in northeast China, the diffusion corresponded to the distribution of railways [51]. Although transportation connectivity had a positive association with plague presence, we did not find a significant association with plague intensity. Local plague intensity therefore might be more dependent on other local conditions (e.g. species richness of commensal rodent plague hosts) than on the introduction of the plague from elsewhere. Alternatively, our measure of transportation connectivity used the network density of the railways and main roads and might miss short-distance diffusion modes of plague transmission [8]. The local intensity of plague may be more strongly correlated with short-distance propagation. As transportation underwent change during the study period, we repeated our analysis using the available transportation routes in the late nineteenth century (electronic supplementary material, figure S8). We found a non-significant positive association between transportation and plague presence ($p < 0.1$), which is not contradictory to our results (electronic supplementary material, text).

## 4.4. Association with precipitation

The higher plague presence and intensity in wet regions than that in dry regions in China might be explained by the resource pulse [56]. High levels of annual precipitation generally correspond to higher levels of primary productivity (pulsed resources) in ecosystems and thus tend to increase the overall rodent abundance in those ecosystems [14]. Increased precipitation can also increase flea population densities through promoting flea survival and reproduction [12]. High rodent and flea population densities in general increase the risk of plague epizootics and thus the transmission potential of plague to human [57].

## 4.5. Implications and limitations

This study takes advantage of existing large datasets of human plague and biodiversity and uses zero-inflated GAMs to analyse the relationship between human plague and host diversity. In summary, we found a positive association between rodent species richness and plague at a large geographical scale. The association strength results also showed that there was an elevated risk of plague presence in grids with high rodent diversity, strengthening the confidence of our findings.

Considering the current distribution of natural plague foci [58], our results suggested that surveillance targeting plague foci in Yunnan, Inner Mongolia and Northwest with high rodent

diversity could be helpful for the prevention of plague, where plague is endemic in rodents and likely to infect humans. Thus, current strategies such as surveillance of rodent abundance, regular serological test of rodents in these foci with high rodent diversity should be taken seriously. It should be pointed out that, although a positive association between diversity and plague presence was found, the protection of biodiversity and control of infectious diseases such as plague was not contradictory considering the frequent association between biodiversity loss and increased disease prevalence [36].

There are some limitations to our study. Evidence shows that not only the number of species correlates with disease risks, the composition and quality of species in a community also play a role [28]. One of our limitations is using the number of species but lack of quantitative measurement of the host quality. Secondly, as the association does not imply causation, more research is needed to study the causal links between diversity and plague, as well as the potential mechanisms behind this association. Finally, deficiencies in the current availability of diversity change data limit our ability to analyse the relationship between diversity change and human plague pattern, which truly needs exploration in future studies.

Ethics. This article does not present research with ethical considerations.
Data accessibility. Data and codes are deposited at the Dryad Digital Repository: https://doi.org/10.5061/dryad.k59bc2m [59].
Authors' contributions. N.C.S., L.X. and Z.S. designed research; Y.X., X.F., S.W., Q.L. and X.W. compiled the data. Z.S., L.X. and B.L. performed research; Z.S., L.X., B.V.S., K.R.D., Z.Z., J.X., B.X. and N.C.S. analysed data; Z.S., L.X., B.V.S., K.R.D. and N.C.S. wrote the paper. All authors gave final approval for publication.
Competing interests. We have no competing interests.
Funding. This research was supported by the National Key Research and Development Program of China (no. 2016YFA0600104), the key grant for international cooperation of the National Science Foundation of China (31420103913), Tsinghua University and the Centre for Ecological and Evolutionary Synthesis (CEES) of the University of Oslo.
Acknowledgement. We thank Dr Anna Mazzaralla for the linguistic revision.

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
