## [Reviewer comments · Royal Society Open Science]

Review History

RSOS-190216.R0 (Original submission)

Review form: Reviewer 1

Is the manuscript scientifically sound in its present form?

No

Are the interpretations and conclusions justified by the results?

No

Is the language acceptable?

No

Is it clear how to access all supporting data?

Yes

Do you have any ethical concerns with this paper?

No

Have you any concerns about statistical analyses in this paper?

No

Recommendation?

Major revision is needed (please make suggestions in comments)

Comments to the Author(s)

Sun et al. "Rodent diversity and other environmental factors affecting the human plague system"

The authors describe the result of a study analyzing the effects of host diversity, transportation connectivity, human population density, and precipitation on the incidence and intensity of plague in China from 1772-1963. Previous work has established that transportation connectivity, precipitation, and human population density affect plague spread and incidence. This study includes various components of vertebrate diversity as predictors, including total vertebrate diversity, rodent diversity, wild rodent diversity, and commensal rodent diversity. They find that rodent diversity is positively correlated with plague incidence and intensity.

This analysis employs several large datasets to address an important disease. The statistical approach is, in principle, acceptable, though its limitations are not as well articulated in the manuscript as they should be (see below). The manuscript follows a common pattern for large-scale analyses as they are being done in ecology these days. The authors find a correlation between two events: when plague occurs, it is associated with high rodent diversity. They then assume that when there is high rodent diversity, there will be plague. This is a fundamental logical fallacy. That relationship would only be true if the connection were causal, but they are far from establishing causality.

The authors could, in principle, test the strength of the association between rodent diversity and plague by asking how frequently high rodent diversity is associated with plague incidence/intensity (the reverse of the association they tested). However, this is a problem since they used all of their plague data to develop the original model. In sum, their approach has fundamental limitations that they do not acknowledge, and their attempt to link their analysis to plague control is premature. If they want to mention plague control, they need to frame that discussion with the steps that would need to be taken to establish a tighter linkage - in particular, a measure of how frequently areas of high rodent diversity are linked with plague outbreaks (the reverse of the association investigated here). They get at the point above in the most preliminary way by noting that their rodent results do not fit the pattern of plague along the Heihe-Tengchong line (lines 301-314). They argue that there must be mitigating factors they do not yet understand, but of course there are other possible explanations and they should take this mismatch between data and predictions much more seriously than they do.

The data do not support the strength of their conclusions. In Table 1, the authors show the percentage of the variance explained by each of their models. The null model for plague intensity, which includes no measures of diversity, explains 24% of the variance. Adding various components of rodent diversity increases that percentage by at most 2%. For plague incidence, the null model explains 15% and adding rodent diversity improves the fit by 4%. The null model doesn't fit the data very well, and adding rodent diversity doesn't change that. I credit the authors with including these measures of the fit, which many authors do not include in model comparisons. However, the small variance explained does not justify the enthusiastic implication that diversity causes plague as in the current manuscript.

Finally, I raise a major concern about the ultimate conclusion they draw: “Our findings help elucidate the role of biotic and socioeconomic factors on human plague cases, suggesting targeted prevention and control measures could be useful in these high-risk regions.” Are they suggesting that rodent control in regions with high rodent diversity would be an appropriate response to their findings? In this form, this statement is unsupported by their data and strikingly irresponsible, with potentially devastating consequences for biodiversity. Indeed, the variance explained suggests that this kind of action, in addition to not being supported by the data, would be largely ineffective and would waste resources that could be better allocated to other mitigation options, thus in one statement inadvertently doing potential damage to both biodiversity and human health.

More minor issues are addressed below:

The manuscript could use a light copy-edit for syntax and spelling (e.g. “adaption”). Author’s numbering Line 23-24 – “however, there is only a limited understanding of the connection between them”. Here, the authors suggest that the connection between biodiversity loss and EIDs is not well understood. That’s actually not the case. We know that the loss of biodiversity frequently leads to increases in infectious diseases (e.g. Civitello et al., which they cite). We also know that ecosystems with higher levels of innate diversity often have higher numbers of pathogens. These are actually fairly well understood. What’s confusing is that some scientists conflate the two phenomena without taking the time to understand the underlying mechanisms that differentiate them. The authors are doing that here, in the Abstract, and again in the Intro and Discussion framing their paper. I encourage them to be much more scholarly around this issue.

Line 72-92 – Here, the authors also need to be more scholarly. They have identified the pattern of innate species diversity leading to higher diversity of pathogens, but they do not seem to understand the dilution effect, despite citing it. This is difficult literature to enter, with some discredited papers in high profile journals, e.g. their reference 26. (See PNAS November 17, 2015 112 (46) E6262). Nevertheless, they do need to understand the mechanisms underlying the phenomena better if they are to invoke the literature here. In their system, variation from location to location in innate biodiversity would not suggest that a dilution effect would operate. However, CHANGE in innate biodiversity would indicate the potential for a dilution effect. If, for example, predators were lost from certain habitats in China and that resulted in increases in abundance of rodents, one could see evidence for a dilution effect. They appear, however, to have no real data on change in their vertebrate communities through time. Thus, what they have is that when you have a higher diversity of good hosts, you get more disease. The mechanisms underlying the dilution effect also rely on variation in host quality. Though they cite papers that show that different species can harbor the plague bacterium, they do not give any evidence that hosts vary in quality, e.g. their potential to transmit the bacterium to feeding vectors. Are hosts all of equal quality here? Again, they are looking at crude correlations with data of limited quality. They need to be clear about what claims they can actually make based on the limited quality and resolution of their data, and they need to be up-front about how limited their data actually are.

Lines 138-139. The authors claim that 1797 species of vertebrates can serve as hosts for plague “or be implicated” but they cite no evidence. I think what they mean is that these species might play some kind of role in the plague disease system, e.g. by regulating host populations, but that this information is not known for most of these species. This distinction needs to be made more clearly. Ideally, Table S2 would indicate a metric of reservoir competence (i.e. a measure of host quality). They do need to indicate how they measure host capacity. My guess is that they use seroprevalence, which is a common but poor measure of a host. Seroprevalence indicates that a host has been exposed to a pathogen, but not that the host can actually transmit it onwards. That

distinction is critically important. If the hosts are exposed but don't transmit the bacteria onward, then they are dead-ends, and we would expect their presence to reduce transmission. If the hosts do transmit it onwards, then we would expect their presence to increase transmission, all else being equal. Without more information on this distinction, they cannot actually establish a priori expectations of outcomes for disease.

Lines 246-252. The authors speculate quite widely about the underlying processes that could underlie their results, despite the fact that their results rely on crude correlations and no understanding of mechanism. The second half of this opening paragraph of the Discussion should be removed from this location and moved to an area of higher speculation later in the Discussion. In addition, it should be bracketed by an exploration of just how little they know about the actual roles of any of these hosts. Essentially, they have gone a little wild with post hoc reasoning here, and are speculating about detailed causal connections that are definitely not warranted by the quality of their data or their analysis.

Line 316. "local population size" should be "local HUMAN population size".

Lines 343-349. The authors mention that a link between precipitation and plague, via a surge in rodent food and then rodent abundance, is an example of a "trophic cascade". However, it is not. A trophic cascade results from a loss of top-down regulation after the disappearance of predators. It results in alternating increases/decreases through a trophic chain. Parmenter et al. misused the concept in their hantavirus work, but that does not support its ongoing misuse here. What these authors actually have is an example of a resource pulse (see Ostfeld et al. 2000 or so), which has a number of important distinctions from a trophic cascade. In a resource pulse, the perturbation is bottom-up and results in increase/increase rather than alternating increase/decrease.

Table 1. The authors present the variance explained by each of their models and try to draw subtle distinctions between them in the text. However, the null model (with no measure of host diversity) explains 24% of the variance and the best model explains only 26%. This is marginal improvement in the fit of the model by any measure and clearly needs to be acknowledged in the main text.

Review form: Reviewer 2 (John Schmidt)

Is the manuscript scientifically sound in its present form?

Yes

Are the interpretations and conclusions justified by the results?

Yes

Is the language acceptable?

Yes

Is it clear how to access all supporting data?

Yes

Do you have any ethical concerns with this paper?

No

Have you any concerns about statistical analyses in this paper?

No

Recommendation?

Accept with minor revision (please list in comments)

Comments to the Author(s)

Overall, I like this study a lot, and I think the findings are very useful. I appreciate the way the authors included reservoir diversity in multiple subsets and the sensitivity of the regression to grid cell size. However, I'm not sure averaging values for human population and transportation over a 200-year period is the best approach (see below).

Major comments

Why average values over the whole period if you are using zero-inflated methods?

Why not include every grid cell in every year from 1772-1964?

You could control for repeated sampling using a geospatial model with the coordinates of the centroid of each grid cell included explicitly in the model.

The last section "Implications for plague prevention" doesn't add much to the manuscript as it is. The work really deserves a better summary with perhaps substantive recommendations in terms of disease surveillance.

Minor comments

P 4, L 88 You may want to reference the dilution effect

P 5, L 118 Do you mean grid cell i? I don't understand the grids very well generally. Is there a grid for each year from 1772-1964? Are these grids then stacked and average intensity calculated for each cell over the period?

P 6, L 125 I don't understand why the human population values are calculated differently for grid (cells?) with and without plague cases.

P 6, L 146 What county boundaries?

P 7, L 160 I don't understand how transportation was summarized over time.

P 9, L 214 "presented a positive effect on" should be "was positively related to"

P 9, L 225 should be "was still the best predictor of"

P 10, L 269 should be "microevolution that accompanies"

P 11, L 284 should be "has been shown"

P 11, L 285 should be "has little effect"

P 12, L 301 should be "distribution of human population"

Decision letter (RSOS-190216.R0)

12-Apr-2019

Dear Dr Stenseth,

The editors assigned to your paper ("Rodent diversity and other environmental factors affecting

the human plague system") have now received comments from reviewers. We would like you to revise your paper in accordance with the referee and Associate Editor suggestions which can be found below (not including confidential reports to the Editor). Please note this decision does not guarantee eventual acceptance.

Please submit a copy of your revised paper before 05-May-2019. Please note that the revision deadline will expire at 00.00am on this date. If we do not hear from you within this time then it will be assumed that the paper has been withdrawn. In exceptional circumstances, extensions may be possible if agreed with the Editorial Office in advance. We do not allow multiple rounds of revision so we urge you to make every effort to fully address all of the comments at this stage. If deemed necessary by the Editors, your manuscript will be sent back to one or more of the original reviewers for assessment. If the original reviewers are not available, we may invite new reviewers.

- Data accessibility

If you wish to submit your supporting data or code to Dryad (<http://datadryad.org/>), or modify your current submission to dryad, please use the following link:
<http://datadryad.org/submit?journalID=RSOS&manu=RSOS-190216>

- Competing interests

- Authors' contributions

- Acknowledgements

- Funding statement

on behalf of Dr John Dalton (Associate Editor) and Kevin Padian (Subject Editor)
openscience@royalsociety.org

Associate Editor's comments (Dr John Dalton):

Associate Editor: 1

Comments to the Author:

The reviewers were positive about you manuscript but one reviewer in particular had substantial and valid suggestions to improve the manuscript. Your attention and response to these would be appreciated.

Editor comments:

With one good and one very thorough review, there are many issues that the authors need to respond to. I considered logging a "reject/resub" decision mainly because our turnaround time for "major revision" is three weeks. What is important here is that the authors respond thoroughly to all issues, because another round of revision will not be possible. SO, if you need more time than provided, please let the editorial office know in advance. Many thanks and best wishes for revising.

Comments to Author:

Reviewers' Comments to Author:

Reviewer: 1

Comments to the Author(s)

Sun et al. "Rodent diversity and other environmental factors affecting the human plague system"

The authors describe the result of a study analyzing the effects of host diversity, transportation connectivity, human population density, and precipitation on the incidence and intensity of plague in China from 1772-1963. Previous work has established that transportation connectivity, precipitation, and human population density affect plague spread and incidence. This study includes various components of vertebrate diversity as predictors, including total vertebrate diversity, rodent diversity, wild rodent diversity, and commensal rodent diversity. They find that rodent diversity is positively correlated with plague incidence and intensity.

This analysis employs several large datasets to address an important disease. The statistical approach is, in principle, acceptable, though its limitations are not as well articulated in the manuscript as they should be (see below). The manuscript follows a common pattern for large-scale analyses as they are being done in ecology these days. The authors find a correlation between two events: when plague occurs, it is associated with high rodent diversity. They then assume that when there is high rodent diversity, there will be plague. This is a fundamental logical fallacy. That relationship would only be true if the connection were causal, but they are far from establishing causality.

The authors could, in principle, test the strength of the association between rodent diversity and plague by asking how frequently high rodent diversity is associated with plague incidence/intensity (the reverse of the association they tested). However, this is a problem since they used all of their plague data to develop the original model. In sum, their approach has fundamental limitations that they do not acknowledge, and their attempt to link their analysis to plague control is premature. If they want to mention plague control, they need to frame that discussion with the steps that would need to be taken to establish a tighter linkage - in particular, a measure of how frequently areas of high rodent diversity are linked with plague outbreaks (the reverse of the association investigated here). They get at the point above in the most preliminary way by noting that their rodent results do not fit the pattern of plague along the Heihe-Tengchong line (lines 301-314). They argue that there must be mitigating factors they do not yet understand, but of course there are other possible explanations and they should take this mismatch between data and predictions much more seriously than they do.

The data do not support the strength of their conclusions. In Table 1, the authors show the percentage of the variance explained by each of their models. The null model for plague intensity, which includes no measures of diversity, explains 24% of the variance. Adding various components of rodent diversity increases that percentage by at most 2%. For plague incidence, the null model explains 15% and adding rodent diversity improves the fit by 4%. The null model doesn't fit the data very well, and adding rodent diversity doesn't change that. I credit the authors with including these measures of the fit, which many authors do not include in model comparisons. However, the small variance explained does not justify the enthusiastic implication that diversity causes plague as in the current manuscript.

Finally, I raise a major concern about the ultimate conclusion they draw: "Our findings help elucidate the role of biotic and socioeconomic factors on human plague cases, suggesting targeted prevention and control measures could be useful in these high-risk regions." Are they suggesting

that rodent control in regions with high rodent diversity would be an appropriate response to their findings? In this form, this statement is unsupported by their data and strikingly irresponsible, with potentially devastating consequences for biodiversity. Indeed, the variance explained suggests that this kind of action, in addition to not being supported by the data, would be largely ineffective and would waste resources that could be better allocated to other mitigation options, thus in one statement inadvertently doing potential damage to both biodiversity and human health.

More minor issues are addressed below:

The manuscript could use a light copy-edit for syntax and spelling (e.g. “adaption”). Author’s numbering Line 23-24 – “however, there is only a limited understanding of the connection between them”. Here, the authors suggest that the connection between biodiversity loss and EIDs is not well understood. That’s actually not the case. We know that the loss of biodiversity frequently leads to increases in infectious diseases (e.g. Civitello et al., which they cite). We also know that ecosystems with higher levels of innate diversity often have higher numbers of pathogens. These are actually fairly well understood. What’s confusing is that some scientists conflate the two phenomena without taking the time to understand the underlying mechanisms that differentiate them. The authors are doing that here, in the Abstract, and again in the Intro and Discussion framing their paper. I encourage them to be much more scholarly around this issue.

Line 72-92 – Here, the authors also need to be more scholarly. They have identified the pattern of innate species diversity leading to higher diversity of pathogens, but they do not seem to understand the dilution effect, despite citing it. This is difficult literature to enter, with some discredited papers in high profile journals, e.g. their reference 26. (See PNAS November 17, 2015 112 (46) E6262). Nevertheless, they do need to understand the mechanisms underlying the phenomena better if they are to invoke the literature here. In their system, variation from location to location in innate biodiversity would not suggest that a dilution effect would operate. However, CHANGE in innate biodiversity would indicate the potential for a dilution effect. If, for example, predators were lost from certain habitats in China and that resulted in increases in abundance of rodents, one could see evidence for a dilution effect. They appear, however, to have no real data on change in their vertebrate communities through time. Thus, what they have is that when you have a higher diversity of good hosts, you get more disease. The mechanisms underlying the dilution effect also rely on variation in host quality. Though they cite papers that show that different species can harbor the plague bacterium, they do not give any evidence that hosts vary in quality, e.g. their potential to transmit the bacterium to feeding vectors. Are hosts all of equal quality here? Again, they are looking at crude correlations with data of limited quality. They need to be clear about what claims they can actually make based on the limited quality and resolution of their data, and they need to be up-front about how limited their data actually are.

Lines 138-139. The authors claim that 1797 species of vertebrates can serve as hosts for plague “or be implicated” but they cite no evidence. I think what they mean is that these species might play some kind of role in the plague disease system, e.g. by regulating host populations, but that this information is not known for most of these species. This distinction needs to be made more clearly. Ideally, Table S2 would indicate a metric of reservoir competence (i.e. a measure of host quality). They do need to indicate how they measure host capacity. My guess is that they use seroprevalence, which is a common but poor measure of a host. Seroprevalence indicates that a host has been exposed to a pathogen, but not that the host can actually transmit it onwards. That distinction is critically important. If the hosts are exposed but don’t transmit the bacteria onward, then they are dead-ends, and we would expect their presence to reduce transmission. If the hosts do transmit it onwards, then we would expect their presence their to increase transmission, all

else being equal. Without more information on this distinction, they cannot actually establish a priori expectations of outcomes for disease.

Lines 246-252. The authors speculate quite widely about the underlying processes that could underlie their results, despite the fact that their results rely on crude correlations and no understanding of mechanism. The second half of this opening paragraph of the Discussion should be removed from this location and moved to an area of higher speculation later in the Discussion. In addition, it should be bracketed by an exploration of just how little they know about the actual roles of any of these hosts. Essentially, they have gone a little wild with post hoc reasoning here, and are speculating about detailed causal connections that are definitely not warranted by the quality of their data or their analysis.

Line 316. "local population size" should be "local HUMAN population size".

Lines 343-349. The authors mention that a link between precipitation and plague, via a surge in rodent food and then rodent abundance, is an example of a "trophic cascade". However, it is not. A trophic cascade results from a loss of top-down regulation after the disappearance of predators. It results in alternating increases/decreases through a trophic chain. Parmenter et al. misused the concept in their hantavirus work, but that does not support its ongoing misuse here. What these authors actually have is an example of a resource pulse (see Ostfeld et al. 2000 or so), which has a number of important distinctions from a trophic cascade. In a resource pulse, the perturbation is bottom-up and results in increase/increase rather than alternating increase/decrease.

Table 1. The authors present the variance explained by each of their models and try to draw subtle distinctions between them in the text. However, the null model (with no measure of host diversity) explains 24% of the variance and the best model explains only 26%. This is marginal improvement in the fit of the model by any measure and clearly needs to be acknowledged in the main text.

Reviewer: 2

Comments to the Author(s)

Overall, I like this study a lot, and I think the findings are very useful. I appreciate the way the authors included reservoir diversity in multiple subsets and the sensitivity of the regression to grid cell size. However, I'm not sure averaging values for human population and transportation over a 200-year period is the best approach (see below).

Major comments

Why average values over the whole period if you are using zero-inflated methods?

Why not include every grid cell in every year from 1772-1964?

You could control for repeated sampling using a geospatial model with the coordinates of the centroid of each grid cell included explicitly in the model.

The last section "Implications for plague prevention" doesn't add much to the manuscript as it is. The work really deserves a better summary with perhaps substantive recommendations in terms of disease surveillance.

Minor comments

P 4, L 88 You may want to reference the dilution effect

P 5, L 118 Do you mean grid cell i? I don't understand the grids very well generally. Is there a grid for each year from 1772-1964? Are these grids then stacked and average intensity calculated for each cell over the period?

P 6, L 125 I don't understand why the human population values are calculated differently for grid (cells?) with and without plague cases.

P 6, L 146 What county boundaries?

P 7, L 160 I don't understand how transportation was summarized over time.

P 9, L 214 "presented a positive effect on" should be "was positively related to"

P 9, L 225 should be "was still the best predictor of"

P 10, L 269 should be "microevolution that accompanies"

P 11, L 284 should be "has been shown"

P 11, L 285 should be "has little effect"

P 12, L 301 should be "distribution of human population"

Author's Response to Decision Letter for (RSOS-190216.R0)

See Appendix A.

Decision letter (RSOS-190216.R1)

21-May-2019

Dear Dr Stenseth,

I am pleased to inform you that your manuscript entitled "Human plague system associated with rodent diversity and other environmental factors" is now accepted for publication in Royal Society Open Science.

on behalf of Dr John Dalton (Associate Editor) and Kevin Padian (Subject Editor)
openscience@royalsociety.org

Follow Royal Society Publishing on Twitter: [@RSocPublishing](https://twitter.com/RSocPublishing)
Follow Royal Society Publishing on Facebook:
<https://www.facebook.com/RoyalSocietyPublishing.FanPage/>
Read Royal Society Publishing's blog: <https://blogs.royalsociety.org/publishing/>

Appendix A

Point-to-point Responses to Reviewers' Comments

Reviewer: 1

Comments to the Author(s)

The authors describe the result of a study analyzing the effects of host diversity, transportation connectivity, human population density, and precipitation on the incidence and intensity of plague in China from 1772-1963. Previous work has established that transportation connectivity, precipitation, and human population density affect plague spread and incidence. This study includes various components of vertebrate diversity as predictors, including total vertebrate diversity, rodent diversity, wild rodent diversity, and commensal rodent diversity. They find that rodent diversity is positively correlated with plague incidence and intensity.

Q1: This analysis employs several large datasets to address an important disease. The statistical approach is, in principle, acceptable, though its limitations are not as well articulated in the manuscript as they should be (see below). The manuscript follows a common pattern for large-scale analyses as they are being done in ecology these days. The authors find a correlation between two events: when plague occurs, it is associated with high rodent diversity. They then assume that when there is high rodent diversity, there will be plague. This is a fundamental logical fallacy. That relationship would only be true if the connection were causal, but they are far from establishing causality.

Response: We fully agree that using association is much better than causality in our study. Thus, we do the following revisions.

(1) We added tests of association strength and significance between rodent diversity and plague, which showed how frequently high diversity and human plague was associated. We used relative risk [1] to evaluate the association strength, and *chi-square* test [2] to examine the association significance (detailed results in next response).

(2) We revised our descriptions across the manuscript, to avoid describing association as causality. We also added a sentence in the last section of 'Discussion' (line number 386 – 388), which stated our limitations as

“Secondly, as association does not imply causation, more research is needed to study the causal links between diversity and plague, as well as the potential mechanisms behind this association.”

Q2: The authors could, in principle, test the strength of the association between rodent diversity and plague by asking how frequently high rodent diversity is associated with

plague incidence/intensity (the reverse of the association they tested). However, this is a problem since they used all of their plague data to develop the original model. In sum, their approach has fundamental limitations that they do not acknowledge, and their attempt to link their analysis to plague control is premature. If they want to mention plague control, they need to frame that discussion with the steps that would need to be taken to establish a tighter linkage – in particular, a measure of how frequently areas of high rodent diversity are linked with plague outbreaks (the reverse of the association investigated here). They get at the point above in the most preliminary way by noting that their rodent results do not fit the pattern of plague along the Heihe-Tengchong line (lines 301-314). They argue that there must be mitigating factors they do not yet understand, but of course there are other possible explanations and they should take this mismatch between data and predictions much more seriously than they do.

Response: To address this issue, we calculated risk ratio (*RR*) [1] and conducted *chi-square* test [2] between two groups, high-diversity and low-diversity grid cells, to measure the strength and significance of association between rodent diversity and plague. We used the average value of gridded species richness of rodent plague hosts in China to classify high-diversity (diversity larger than the average) and low-diversity groups (diversity less than or equal to the average). Detailed methods and results were added in the main text (lines 203 – 208, 238 – 244) and SI (Table S3 and S4, lines 130 – 155 in SI). Key points included:

(1) We calculated *RR*, the ratio of plague presence in high-diversity group to low-diversity group, as a measure of the association strength. The frequency of plague presence (plague have occurred) and absence (plague never occurred) in high/low diversity grid cells was shown in Table S3.

Table S3. The observational frequency of plague presence in high-diversity and low-diversity grid cells.

	Presence of plague (number of grids)	Absence of plague (number of grids)	Sum	Ratio of plague presence
High diversity	158	331	489	32.31%
Low diversity	81	549	630	12.86%
Sum	239	880	1119	21.36%

Based on Table S3, the ratio of plague presence in high-diversity group (R_h) was 32.31%, while the ratio in low-diversity group (R_l) was 12.86%. The overall ratio (R) was 21.36% for all grids. Using ‘*fmsb*’ package in R, *RR* was 2.51 (95%CI: 1.98–3.20; $p < 0.01$), which

indicated that grids with high diversity of rodent plague hosts had 2.51 times the risk of plague presence compared to grids with low diversity.

(2) We tested the association significance using *chi-square* test. The null hypothesis (H_0) was that plague presence is not associated with rodent diversity. If the null hypothesis is true, the expected counts in each category (Table S4) calculated by using the overall ratio R (21.36%) would be close to the observed values. Otherwise, the null hypothesis is not valid under large χ^2 .

Table S4. The expected frequency of plague presence in high-diversity and low-diversity grid cells.

	Presence of plague (number of grids)	Absence of plague (number of grids)
High diversity	104.45	384.55
Low diversity	134.57	495.43

Using *chisq.test* function in R, χ^2 was 60.88 ($d.f. = 1, p < 0.01$). Thus, the null hypothesis is rejected. The result indicated a significant association between high rodent diversity and plague presence.

Q3: The data do not support the strength of their conclusions. In Table 1, the authors show the percentage of the variance explained by each of their models. The null model for plague intensity, which includes no measures of diversity, explains 24% of the variance. Adding various components of rodent diversity increases that percentage by at most 2%. For plague incidence, the null model explains 15% and adding rodent diversity improves the fit by 4%. The null model doesn't fit the data very well, and adding rodent diversity doesn't change that. I credit the authors with including these measures of the fit, which many authors do not include in model comparisons. However, the small variance explained does not justify the enthusiastic implication that diversity causes plague as in the current manuscript.

Response: This comment is the same as the last comment in “Minor issues”, which requires a clearer description of the model improvement, and also a clearer expression of association in the manuscript (not causality). We made the following revisions.

(1) We added a sentence to indicate the marginal improvement in the model fit in “Results” in the main text (lines 226 – 229) as below,

“Model (v) gave a marginal improvement of the model performance, with a 1.5 to 4.3 percent-point increase in the variance explained of the presence of plague, to a maximum

of 19.2%, and a 0.7 to 2.2 percent-point increase to a maximum of 26.3% of the variance explained of the intensity of plague outbreaks (Table 1).”

(2) We revised our description of the diversity-plague relationship. We changed “species richness present a positive effect on plague” to “species richness was positively associated with plague” (lines 230 - 231 and 249 - 250).

Q4: Finally, I raise a major concern about the ultimate conclusion they draw: “Our findings help elucidate the role of biotic and socioeconomic factors on human plague cases, suggesting targeted prevention and control measures could be useful in these high-risk regions.” Are they suggesting that rodent control in regions with high rodent diversity would be an appropriate response to their findings? In this form, this statement is unsupported by their data and strikingly irresponsible, with potentially devastating consequences for biodiversity. Indeed, the variance explained suggests that this kind of action, in addition to not being supported by the data, would be largely ineffective and would waste resources that could be better allocated to other mitigation options, thus in one statement inadvertently doing potential damage to both biodiversity and human health.

Response: We thank the reviewer for pointing out this issue. The reviewer 2 also suggested that this part needs a better summary. We changed this section to a summary of this study and its potential application, as well as limitations.

We made recommendations for plague surveillance with the consideration of natural plague foci distribution in China (there is no natural plague foci in the central part of China) [3] and results of association strength between high diversity and plague presence. Details are as follows (lines 368 – 382):

“This study takes advantage of existing large datasets of human plague and biodiversity, and uses zero-inflated GAMs to analyze the relationship between human plague and host diversity. In summary, we found a positive association between rodent species richness and plague at a large geographic scale. The association strength results also showed that there was an elevated risk of plague presence in grids with high rodent diversity, strengthening the confidence of our findings.

Considering the current distribution of natural plague foci [3], our results suggested that surveillance targeting plague foci in Yunnan, Inner Mongolia and Northwest with high rodent diversity could be helpful for the prevention of plague, where plague is endemic in rodents and likely to infect humans. Thus, current strategies such as surveillance of rodent abundance, regular serological test of rodents in these foci with high rodent diversity

should be taken seriously. It should be pointed out that, although a positive association between diversity and plague presence was found, the protection of biodiversity and control of infectious diseases such as plague was not contradictory considering the frequent association between biodiversity loss and increased disease prevalence [4].”

We added limitations of this study in the last paragraph of this section (lines 383 - 390), as below

“There are some limitations to our study. Evidence shows that not only the number of species correlates with disease risks, the composition and quality of species in a community also play a role [5]. One of our limitations is using the number of species but lack of quantitative measurement of the host quality. Secondly, as association does not imply causation, more research is needed to study the causal links between diversity and plague, as well as the potential mechanisms behind this association. Finally, deficiencies in current availability of diversity change data limit our ability to analyze the relationship between diversity change and human plague pattern, which truly needs exploration in future studies.”

More minor issues are addressed below:

Q5: The manuscript could use a light copy-edit for syntax and spelling (e.g. “adaption”).

Response: Thanks for this suggestion. Proofreading was performed by native English speakers in the revised manuscript.

Q6: Author’s numbering Line 23-24 – “however, there is only a limited understanding of the connection between them”. Here, the authors suggest that the connection between biodiversity loss and EIDs is not well understood. That’s actually not the case. We know that the loss of biodiversity frequently leads to increases in infectious diseases (e.g. Civitello et al., which they cite). We also know that ecosystems with higher levels of innate diversity often have higher numbers of pathogens. These are actually fairly well understood. What’s confusing is that some scientists conflate the two phenomena without taking the time to understand the underlying mechanisms that differentiate them. The authors are doing that here, in the Abstract, and again in the Intro and Discussion framing their paper. I encourage them to be much more scholarly around this issue.

Line 72-92 – Here, the authors also need to be more scholarly. They have identified the pattern of innate species diversity leading to higher diversity of pathogens, but they do not seem to understand the dilution effect, despite citing it. This is difficult literature to enter, with some discredited papers in high profile journals, e.g. their

reference 26. (See PNAS November 17, 2015 112 (46) E6262). Nevertheless, they do need to understand the mechanisms underlying the phenomena better if they are to invoke the literature here. In their system, variation from location to location in innate biodiversity would not suggest that a dilution effect would operate. However, CHANGE in innate biodiversity would indicate the potential for a dilution effect. If, for example, predators were lost from certain habitats in China and that resulted in increases in abundance of rodents, one could see evidence for a dilution effect. They appear, however, to have no real data on change in their vertebrate communities through time. Thus, what they have is that when you have a higher diversity of good hosts, you get more disease. The mechanisms underlying the dilution effect also rely on variation in host quality. Though they cite papers that show that different species can harbor the plague bacterium, they do not give any evidence that hosts vary in quality, e.g. their potential to transmit the bacterium to feeding vectors. Are hosts all of equal quality here? Again, they are looking at crude correlations with data of limited quality. They need to be clear about what claims they can actually make based on the limited quality and resolution of their data, and they need to be up-front about how limited their data actually are.

Response: We thoroughly revised the Abstract, Introduction and Discussion accordingly. We also added “dilution effect” in Introduction. Detailed revisions are shown as below.

(1) In the abstract, we revised the background of this study (lines 22 - 25) to

“Plague remains a threat to public health and is considered as a re-emerging infectious disease today. Rodents play an important role as major hosts in plague persistence and driving plague outbreaks in natural foci; however, few studies have tested the association between host diversity in ecosystems and human plague risk.”

(2) In the Introduction, together with the comments from Reviewer 2, we reframed this section by adding the ‘dilution effect’ (lines 80 – 81, lines 86 – 91). References were also adjusted. Detailed revisions were

“The human infectious diseases are affected by both the local innate species richness and their changes [6-10]. Regions with high species richness are likely to be source pool for new pathogens [8]. Species richness has been shown to have a positive association with the emergence of zoonotic diseases [7,10] and the species richness of human pathogens [9]. High innate species richness may provide a stable system for pathogen maintenance and transmission, and also prevent local extinction of pathogens, which in turn affects human health [9]. However, the loss of biodiversity frequently leads to increases in infectious diseases, known as the dilution effect [6]. Dilution effect often occurs in disease systems in which hosts differ in competence for transmission of pathogens and the

competent host populations increase as the biodiversity declines [5]. Instead, high diversity with non-competent hosts tends to lower the abundance of competent hosts and reduces the disease risk [5,6].”

(3) In Discussion, we removed Salkeld *et al.* [11] and added a reference of Bonvicino *et al.* [12], which showed that natural plague foci in South America had a higher rodent diversity (lines 275 - 276).

(4) Hosts in the assemblage of “known rodent plague hosts in China” (Table S2) were considered to be of equal quality here (also see our next response related to the measure of host capacity). The limitations related to measure of host capacity of our diversity data were added in main text in lines 383 – 386.

Q7: Lines 138-139. The authors claim that 1797 species of vertebrates can serve as hosts for plague “or be implicated” but they cite no evidence. I think what they mean is that these species might play some kind of role in the plague disease system, e.g. by regulating host populations, but that this information is not known for most of these species. This distinction needs to be made more clearly. Ideally, Table S2 would indicate a metric of reservoir competence (i.e. a measure of host quality). They do need to indicate how they measure host capacity. My guess is that they use seroprevalence, which is a common but poor measure of a host. Seroprevalence indicates that a host has been exposed to a pathogen, but not that the host can actually transmit it onwards. That distinction is critically important. If the hosts are exposed but don’t transmit the bacteria onward, then they are dead-ends, and we would expect their presence to reduce transmission. If the hosts do transmit it onwards, then we would expect their presence to increase transmission, all else being equal. Without more information on this distinction, they cannot actually establish a priori expectations of outcomes for disease.

Response: We have made the following revisions,

(1) We added citations (Gage and Kosoy [13] and Wang *et al.* [14]) which showed the potential role of mammals and birds on plague. We also revised “be implicated in plague” to (in lines 138 - 141):

“There are 1797 species of terrestrial vertebrates recorded in the database, which potentially serve as plague hosts or play an indirect role in plague through predation or competition relations with plague hosts [13,14]. We started from this assemblage, consistent with previous studies of diversity-disease relationship [4,9].”

(2) The measurement of host capacity in Table S2 was not only based on seroprevalence, but also field surveys and bacteriological examinations. We added the measurement of host capacity in lines 145 – 146 in main text and lines 55 – 65 in SI.

There are 88 species of known plague hosts in China based on field surveys of naturally infected animals and etiological investigations [14-16]. The plague hosts are identified based on field surveys of naturally infected animals after 1945, with the criteria of plague infection existing in the host populations [16]. After the field surveys, animal serological examinations (seroprevalence) and bacteriological examinations (detection of *Yersinia pestis*) were both conducted to confirm plague hosts. There are plenty of literatures related to this issue in China, e.g. Wu *et al.* [17], Li *et al.* [18], Ping [19] and Liu [20]. By combining field surveys and etiological studies, these 88 species are thought to be involved in plague ecosystems and transmission in China, in other words, they have the ability to transmit plague to human [16]. In the main text, we stated that (lines 143 – 147):

“Additionally, there are 88 species of known plague hosts in China which are involved in plague ecosystems and transmission (“plague hosts” for short) based on field surveys of naturally infected animals and etiological studies (electronic supplementary material, Text), with 85 species in Mammalia and 3 species in Aves [14].”

Q8: Lines 246-252. The authors speculate quite widely about the underlying processes that could underlie their results, despite the fact that their results rely on crude correlations and no understanding of mechanism. The second half of this opening paragraph of the Discussion should be removed from this location and moved to an area of higher speculation later in the Discussion. In addition, it should be bracketed by an exploration of just how little they know about the actual roles of any of these hosts. Essentially, they have gone a little wild with post hoc reasoning here, and are speculating about detailed causal connections that are definitely not warranted by the quality of their data or their analysis.

Response: These sentences were removed and merged with the later part of “Association with species richness” in Discussion (lines 302 to 306). Besides, we added limitations in lines 383 to 386, which related to our inadequate understanding of the actual roles of each host species.

Q9: Line 316. “local population size” should be “local HUMAN population size”.

Response: Revised as suggested.

Q10: Lines 343-349. The authors mention that a link between precipitation and plague, via a surge in rodent food and then rodent abundance, is an example of a “trophic cascade”. However, it is not. A trophic cascade results from a loss of top-down regulation after the disappearance of predators. It results in alternating increases/decreases through a trophic chain. Parmenter et al. misused the concept in their hantavirus work, but that does not support its ongoing misuse here. What these authors actually have is an example of a resource pulse (see Ostfeld et al. 2000 or so), which has a number of important distinctions from a trophic cascade. In a resource pulse, the perturbation is bottom-up and results in increase/increase rather than alternating increase/decrease.

Response: We revised to “resource pulse” to explain the association between precipitation and human plague (lines 360 - 363). We also added reference of Ostfeld and Keesing [21].

Q11: Table 1. The authors present the variance explained by each of their models and try to draw subtle distinctions between them in the text. However, the null model (with no measure of host diversity) explains 24% of the variance and the best model explains only 26%. This is marginal improvement in the fit of the model by any measure and clearly needs to be acknowledged in the main text.

Response: We shared the reviewer’s concern and added a sentence to state this marginal improvement in lines 226 to 229 in the main text.

Reference:

1. Porta M. 2014 *Dictionary of epidemiology*. pp. 245-252. Oxford, England, Oxford University Press.
2. Fisher R.A. 1922 On the Interpretation of χ^2 from Contingency Tables, and the Calculation of P. *J R Stat Soc Series B Stat Methodol* **85**(1), 87-94. (doi:10.2307/2340521).
3. Fang X., Xu L., Liu Q., Zhang R. 2011 Eco-geographic landscapes of natural plague foci in China I. Eco-geographic landscapes of natural plague foci. *Chin J Epidemiol* **32**(12), 1232-1236.
4. Morand S., Jittapalapong S., Suputtamongkol Y., Abdullah M.T., Tan B.H. 2014 Infectious Diseases and Their Outbreaks in Asia-Pacific: Biodiversity and Its Regulation Loss Matter. *PLoS One* **9**(2), e90032. (doi:10.1371/journal.pone.0090032).
5. Ostfeld R.S., Keesing F. 2012 Effects of host diversity on infectious disease. *Annu Rev Ecol Evol Syst* **43**, 157-182. (doi:10.1146/annurev-ecolsys-102710-145022).
6. Keesing F., Holt R.D., Ostfeld R.S. 2006 Effects of species diversity on disease risk. *Ecol Lett* **9**(4), 485-498. (doi:10.1111/j.1461-0248.2006.00885.x).
7. Jones K.E., Patel N.G., Levy M.A., Storeygard A., Balk D., Gittleman J.L., Daszak P. 2008 Global trends in emerging infectious diseases. *Nature* **451**(7181), 990-993. (doi:10.1038/nature06536).
8. Keesing F., Belden L.K., Daszak P., Dobson A., Harvell C.D., Holt R.D., Hudson P., Jolles A., Jones K.E., Mitchell C.E. 2010 Impacts of biodiversity on the emergence and transmission of infectious diseases. *Nature* **468**(7324), 647-652. (doi:10.1038/nature09575).

9. Dunn R.R., Davies T.J., Harris N.C., Gavin M.C. 2010 Global drivers of human pathogen richness and prevalence. *Proc Biol Sci* **277**(1694), 2587-2595. (doi:10.1098/rspb.2010.0340).
10. Allen T., Murray K.A., Zambrana-Torres C., Morse S.S., Rondinini C., Marco M.D., Breit N., Olival K.J., Daszak P. 2017 Global hotspots and correlates of emerging zoonotic diseases. *Nat Commun* **8**(1), 1124. (doi:10.1038/s41467-017-00923-8).
11. Salkeld D.J., Padgett K.A., Jones J.H. 2013 A meta-analysis suggesting that the relationship between biodiversity and risk of zoonotic pathogen transmission is idiosyncratic. *Ecol Lett* **16**(5), 679-686. (doi:10.1111/ele.12101).
12. Bonvicino C.R., Oliveira J.A., Cordeiro-Estrela P., D'Andrea P.S., Almeida A.M. 2015 A Taxonomic Update of Small Mammal Plague Reservoirs in South America. *Vector Borne Zoonotic Dis* **15**(10), 571-579. (doi:10.1089/vbz.2015.1788).
13. Gage K.L., Kosoy M.Y. 2005 Natural history of plague: perspectives from more than a century of research. *Annu Rev Entomol* **50**, 505-528. (doi:10.1146/annurev.ento.50.071803.130337).
14. Wang Y., Liu Q., Cong X., Xu C., Li Y. 2007 Plague reservoirs and their classification in natural foci of China. *Chin J Vector Biol Control* **18**(2), 127-133.
15. Fang X. 1990 *Natural foci of plague in China*. pp. 1-290. Beijing, People's Medical Publishing House.
16. Qin C., Xu L., Zhang R., Liu Q., Li G., Fang X. 2012 Ecological-geographic landscapes of natural plague loci in China V. Biological characteristics of major natural reservoirs of *Yersinia pestis*. *Chin J Epidemiol* **33**(8), 692-697.
17. Wu L., Chen Y.H., Pollitzer R., Wu C.Y. 1936 *Plague: a Manual for Medical and Public Health Workers*. pp. 1-421. Shanghai, Weishengshu National Quarantine Service.
18. Li C., Zheng Y., Wang H., Wang Z., Wang L., Wei S., Cui B., Wang G., Chen H., Qi Z., et al. 2009 Investigation on the space structure of plague natural foci in the Sanjiangyuan area in Qinghai Province. *Chin J Endemiol* **28**(5), 522-526.
19. Ping C. 2013 Analysis of plague monitoring results in Lhasa, Tibet during 1988 and 2011. *Chin J Endemiol* **32**(4), 452-452.
20. Liu Y. 2000 *The plague and environment atlas of China*. Beijing, Science Press.
21. Ostfeld R.S., Keesing F. 2000 Pulsed resources and community dynamics of consumers in terrestrial ecosystems. *Trends Ecol Evol* **15**(6), 232-237. (doi:10.1016/S0169-5347(00)01862-0).

Reviewer: 2

Comments to the Author(s)

Overall, I like this study a lot, and I think the findings are very useful. I appreciate the way the authors included reservoir diversity in multiple subsets and the sensitivity of the regression to grid cell size. However, I'm not sure averaging values for human population and transportation over a 200-year period is the best approach (see below).

Major comments

Q1: Why average values over the whole period if you are using zero-inflated methods?

Why not include every grid cell in every year from 1772-1964?

Response: We used average number of plague cases in grids as plague intensity because time-series data of diversity and transportation were not available over the study period during 1772 and 1964. To avoid the pseudo-replication problem [1] caused by including every grid cell in every year from 1772 to 1964, we used a spatial GAM to test the association between diversity and human plague risk. Considering that the first appearance of plague differed among different grids (e.g. plague in Yunnan started from 1772 while plague in Guangxi first appeared in 1867) [2], the human plague cases were averaged as a measurement of plague intensity.

The average plague intensity was still zero-inflated as shown in Fig. S1 in SI. Thus we constructed a spatial zero-inflated GAM.

We added some sentences to explain why we used a spatial GAM in “Zero-Inflated Generalized Additive Model” in the section of ‘Materials and Methods’ (lines 173 – 176) as

“As time-series data of species richness and transportation connectivity were not available, we used a spatial generalized additive model to analyze the association between human plague and environmental factors, in order to avoid the pseudo-replication problem [1] caused by including every grid cell in every year from 1772 to 1964.”

Q2: You could control for repeated sampling using a geoaddivitive model with the coordinates of the centroid of each grid cell included explicitly in the model.

Response: We did not include coordinates of the grid centroids because:

(1) The aim of this study is to investigate the relationship between spatial variation of host diversity and human plague. The smooth function of geographical coordinates can be considered as other variables that are related to the geographical locations. Thus, adding this term would affect our results.

(2) Adding the smooth function of geographical coordinates (thin plate spline) would definitely add the total degrees of freedom to the model. However, the actual meaning of this term was often hard to understand and explain.

Q3: The last section “Implications for plague prevention” doesn’t add much to the manuscript as it is. The work really deserves a better summary with perhaps substantive recommendations in terms of disease surveillance.

Response: We fully agreed with this comment. We changed the last section to a better summary of this study with its potential applications and limitations.

As the reviewer 1 suggested, we added an association strength test of high diversity and plague presence before we made recommendations for plague surveillance. Besides, we also considered the distribution of current natural plague foci in China [3]. There is no natural plague foci in Central China. However, part of the central region near the Heihe-Tengchong line has a high rodent diversity. In order to avoid waste of resources, we make recommendations for plague surveillance with consideration of natural plague foci distribution. Finally, this paragraph has been revised accordingly (lines 368 – 382).

We also added a few sentences to indicate the limitations of this study in lines 383 to 390.

Minor comments

Q4: P4, L88 You may want to reference the dilution effect.

Response: We added the association between diversity change and disease risk (the dilution effect) in lines 86 – 91.

Q5: P5, L118 Do you mean grid cell i ? I don’t understand the grids very well generally. Is there a grid for each year from 1772-1964? Are these grids then stacked and average intensity calculated for each cell over the period?

Response: Yes, we mean grid cell i . As we have explained in response to Q1 in the ‘Major comments’, we used a spatial GAM to test the association between diversity and human plague risk in China to avoid pseudo-replication problem [1]. Thus, each grid cell appears only once in the data of the model. The total number of grids in the model is 1,119 under the one-degree grids.

We added a few sentences in the materials and methods, to make it clearer to readers (lines 173 – 176).

Q6: P6, L125 I don't understand why the human population values are calculated differently for grid (cells?) with and without plague cases.

Response: We did not use the average population over the study period for all grids for two reasons:

(1) The first appearance of plague differed among different grids [2] and plague outbreaks were often discontinuous, with years of stasis in recorded cases [4]. We adjust the population for grids with plague in case there is a huge change in the gridded population from 1772 to 1964.

(2) We also tested and compared the results of using average population for all grids and the adjusted values. We get a better BIC (Bayesian information criterion, accounting by logE [5] in zero-inflated GAM) when we used the different population for grids with and without cases.

These reasons were shown in the SI, in lines 34 to 40.

Q7: P6, L146 What county boundaries?

Response: The species richness used in the model was calculated from a county-level species distribution dataset [6]. Suppose that a specific species was found to distribute in county 'a', we then conducted an overlapping analysis between county 'a' (surface data) and the one-degree grids (gridded data) in China, to determine the presence or absence of the specific species in a grid by 1 or 0. By repeating the above procedure, we can calculate species richness for different subsets.

Our previous description is somewhat confusing, thus we revised this sentence (lines 150 - 152) to

“We overlaid the county-level spatial distribution of each species (polygons) and the one-degree grids covering China, and determined the presence or absence of a specific species in a grid by 1 or 0”.

Q8: P7, L160 I don't understand how transportation was summarized over time.

Response: As time-series data of historical transportation distribution during 1772 and 1964 was not available, we did not use average values of transportation. We only have two historical transportation maps in late 19th century and the first half of the 20th century. So we digitalized these two maps to obtain historical railways and main roads (figure 1B in the main text and figure S8 in SI).

We first used the transportation connectivity in the 20th century to test the association between transportation and human plague. Then we used transportation connectivity in the late 19th century to test the robustness of transportation in the model. This kind of robustness test by using transportation data from different times has also been used in Xu *et al.* [2], where they analysed the association between presence of transportation and the velocity of plague spread.

This method was shown in the section of ‘Transportation and precipitation’ within the Methods in the manuscript (lines 160 - 168), and was also shown in SI in lines 182 to 186. We also illustrated this robustness test result in the section of “Association with human population and transportation connectivity” in Discussion (lines 353 - 358).

Q9: P9, L214 “presented a positive effect on” should be “was positively related to”
P9, L225 should be “was still the best predictor of”

Response: Revised as suggested.

Q10: P10, L269 should be “microevolution that accompanies”

Response: Revised as suggested. In order to avoid such mistakes, proofreading was performed by native English speakers in the revised manuscript.

Q11: P11, L284 should be “has been shown”
P11, L285 should be “has little effect”
P12, L301 should be “distribution of human population”

Response: Revised as suggested.

Reference

1. Heffner R.A., Butler M.J., Reilly C.K. 1996 Pseudoreplication Revisited. *Ecology* **77**(8), 2558-2562. (doi:10.2307/2265754).
2. Xu L., Stige L.C., Kausrud K.L., Ari T.B., Wang S., Fang X., Schmid B.V., Liu Q., Stenseth N.C., Zhang Z. 2014 Wet climate and transportation routes accelerate spread of human plague. *Proc Biol Sci* **281**(1780), 20133159. (doi:10.1098/rspb.2013.3159).
3. Fang X., Xu L., Liu Q., Zhang R. 2011 Eco-geographic landscapes of natural plague foci in China I. Eco-geographic landscapes of natural plague foci. *Chin J Epidemiol* **32**(12), 1232-1236.
4. Institute of Epidemiology and Microbiology, Chinese Academy of Medical Sciences. 1980 *The History of Spread of Plague in China*. pp. 33-1794. Beijing, People's Medical Publishing House.

5. Liu H., Chan K.-S. 2010 Introducing COZIGAM: an R package for unconstrained and constrained zero-inflated generalized additive model analysis. *J Stat Softw* **35**(11), 1-26. (doi:10.18637/jss.v035.i11).
6. Xie Y., Zhang S., Wang W. 2009 *Geographical Atlas of Biodiversity in China*. pp. 1-246. Changsha, Hunan Education Press.